

# Retrieval of health-related UV doses from PAR measurements

Marcelo de Paula Corrêa[1], Sophie Godin-Beekmann[2], Fabrina Bolzan Martins[1], Kátia Mendes[1], Martial Haeffelin[3], Miguel Rivas[4], Elisa Rojas[4]

[1] Instituto de Recursos Naturais, Universidade Federal de Itajubá, Itajubá/MG, Brazil
[2] LATMOS/OVSQ, Université de Versailles Saint-Quentin-en-Yvelines, Guyancourt, France
[3] LMD/IPSL – Ecole Polytechnique, Palaiseau, France
[4] Laboratório de Radiación Solar Ultravioleta, Facultad de Ciências, Universidad de Tarapacá. Arica, Chile

**Correspondence to**: Marcelo de Paula Corrêa (mpcorrea@unifei.edu.br)

**Abstract.** The ultraviolet index (UVI) is an important tool to raise public awareness of the risks of solar radiation overexposure. However, solar ultraviolet radiation instruments are relatively expensive, and the deployment and maintenance of a large-scale network proved impractical. In this paper, we describe a simple method for UVI and daily erythemal dose retrieval from photosynthetically active radiation (PAR – 0.40-0.70 μm) measurements. These latter are generally performed by cheaper instruments and commonly found in any ordinary meteorological station. Despite the different interactions involving UV and PAR radiation with the atmospheric components, our method is based on the solar position and cloud modification factor (CMF) that exert a quite similar influence on these electromagnetic radiation bands. We also show that this method is not suitable for shortwave radiation sensors (0.25-4.00 μm). A large dataset was used to test this method and the comparisons between UVI measurements and inferences are comparable to the instrumental errors. Thus, in this paper we show a 2nd degree regression equation for obtaining erythemal UV from PAR measurements.

## 1 Introduction

Solar ultraviolet radiation (UVR) overexposure is the most important risk factor for skin cancers (Armstrong and Kricker, 2001). On the other hand, a regular sun exposure is required for the synthesis of vitamin D and the prevention of several diseases (Grant, 2006). Due to these important issues, the World Health Organization recommends the ultraviolet index (UVI) as the tool to promote an adequate behavior of sun exposure and to reduce the skin cancer incidence around the world (WHO, 2002). The UVI is a dimensionless scale directly related to the UVR weighted by the spectral response for the reddening of the human skin, named erythemal UVR (E-UVR) (McKinlay and Diffey, 1987). This scale ranges from zero upward and the higher the index value, the greater the potential for harmful effects to the human health. Nevertheless, UVI is not only used to evaluate this harmful effect but it also may be used to estimate the vitamin D production related to the sun exposure (Fioletov et al., 2009; McKenzie et al., 2009). Thus, E-UVR measurements are not only essential to report the UVI to the people, but also to several scientific purposes.



The instruments designated for E-UVR measurements must be quite sensitive and requires ongoing maintenance; consequently, they are usually expensive and the implementation of comprehensive networks makes unviable. Low cost "non-scientific" instruments can be found in ordinary shops, but these instruments are inefficient to perform reliable measurements (Corrêa et al. 2010). On the other hand, satellites could be an information source for large spatial scale

networks. However, the inference of E-UVR from remote sensing measurements shows large uncertainties, mainly under cloudy conditions (Jégou et al., 2011). Thus, E-UVR monitoring over large areas should be complemented by mathematical estimates provided by radiative transfer models; but the effects of clouds, aerosols and ozone short term variability are invariably underestimated in this case. Weather stations usually have instruments to measure solar radiation and they could provide a real picture of the surface radiation levels. However, such radiometers do not always provide E-UVR

measurements.

McKenzie et al. (2004) showed a relationship between E-UVR and UVB radiation broadband measurements, but there is no simple conversion between both terms because they depend on the solar zenith angle (SZA) and ozone amount. Anyway, this problem can be circumvented because SZA is easily calculated and the total ozone content can be obtained by satellite measurements with good accuracy. On the other hand, cloudiness is not yet well modeled and the temporal changes of cloud

cover enhance these uncertainties. However, simple cloud transmission functions can be used to derive the cloud radiative effects mainly over the photosynthetically active radiation (PAR) and UVR spectral regions. These spectral regions are not affected by the water vapor and the radiation scattering is the solely responsible for the radiation attenuation by clouds (Alados et al., 2000). In association with large datasets, this kind of evaluation can provide a better knowledge about the cloudiness influence on the local PAR and UVR variability.

The cloud effect can also be evaluated by the ratio between the solar measurements and the clear-sky calculations, named cloud modification factor (CMF) (Calbo et al., 2005). Schwander et al. (2002) showed that the CMF parameterization is able to provide a good estimation of actual CMFs, also for places with different cloud climatology. It is well-known that the cloud cover decreases UVR transmission to a lesser degree than transmission in the visible spectral range. Thus, the relationship between CMF for solar radiation and the CMF for UVR is not straight forward, but the first one has proved to be a solid

basis to derive this factor for the UVR (Staiger et al., 2008). More recently, Badosa et al. (2014) introduced new methods for retrieving UVI from shortwave radiation (SWR) measurements and cloud recognition by a sky imaging instrument. Despite the suitable estimates provided by these methods, they are more complex and sky imagers are very expensive and not usually found in meteorological stations.

Since UVR and PAR calculations generally show good accuracy under clear-sky conditions and the cloud cover influence

can be evaluated by the relationship of the CMF at each spectral band, we propose in this paper to retrieve UVI and erythemal daily doses from PAR broadband measurements. We have tested our method using the PAR measurements provided by instruments usually found in the most common weather stations. This fact consists of the main motivation of this work, because these instruments are about an order of magnitude cheaper than the UV radiometers; and the available databases are much larger than UVI data. We have used three robust datasets collected in sites with quite distinct



geographical and climate characteristics: Paris, France; São Paulo, Brazil; and Arica, Chile. The results show that this method is effective not only for estimating UVI but also to estimate the cumulative doses on days with or without cloudiness. On the other hand, we also have tested the same method using SWR measurements. However, as expected, the method did not provide accurate results. Thus, the PAR-UVI technique can be an alternative to the lack of special instrumentation and it allows mapping the variability of UVI from the conventional meteorological networks.

This paper is organized as follows: instruments, theoretical methods and dataset information are briefly described in the next section. Results, statistical analysis and discussion are presented in section 3. The last section shows the conclusions and discusses the perspectives on comparisons with different data collected in Europe and South America.

## 2. Methods

First of all, we analyze three large databases of UV, PAR and SWR measurements with the following characteristics showed in the Table 1:

**Table 1.** Information about the measurements used in this study

| Site | *Site Instrumental de Recherche par Télédétection Atmosphérique* (SIRTA) | *Universidade de São Paulo* (USP) | *Universidad de Tarapacá* (UTA) |
|---|---|---|---|
| Local | Paris, France | São Paulo, Brasil | Arica, Chile |
| Latitude | 48.7°N | 21.5°S | 18.5°S |
| Longitude | 2.2°E | 46.5°W | 70.3°W |
| Altitude (m) | ~54 | ~850 | Sea level |
| PAR instrument | Kipp&Zonen Quantum Sensor | Li-cor LI-190 quantum sensor | Biospherical Instr. Inc. GUV-2511 |
| SWR instrument | Li-cor LI-200R Pyranometer | Li-cor LI-200R Pyranometer | – |
| UV instrument | Kipp&Zonen UVS-AE-T radiometer | Solarlight UVB501 biometer | Solarlight UVB501 biometer |
| Period | Apr/2012 – Dec/2013 | Jun/2005 – Jun/2010 | Jan/2011 – Dec/2011 |
| Data record | 1 min | 10 min | 5 min |
| Sample size (datapoints) | ~ 300,000 | ~ 400,000 | ~ 35,000 |

From the French and Brazilian databases, we calculate the CMF for measurement values with the following equation:

$$CMF_R = \frac{R}{R_o} \tag{1}$$



Where R is the broadband measurement (UV, PAR or SWR) and $R_o$ is the clear-sky calculation provided by a radiative transfer model. In this work, these calculations were performed using the TUV (Madronich, 1992) and SBDART (Ricchiazzi et al., 1998) radiative transfer codes. In our calculations, we have considered standard atmosphere profiles, constant background aerosol amounts and total ozone contents were provided by OMI/NASA databases (http://omi.ozoneaq.nasa.gov).

These large CMF samples provided a relevant characterization of the daily and seasonal solar radiation behavior under different cloudiness in each region. Then, from an UVI clear-sky calculation (UVIo) we estimate the UVI through this 2nd order polynomial regression:

$$UVI_{est} = UVI_o \left[ b_0 + b_1 \cdot CMF_{PAR} + b_2 \cdot CMF_{PAR}^2 \right] \tag{2}$$

Where $CMF_{PAR} = PAR/PAR_o$; and, $b_0$, $b_1$ and $b_2$ are the coefficients of the polynomial regression between the series of $CMF_{UVI}$ and $CMF_{PAR}$. Similarly, we also estimate UVI using $CMF_{SWR}$ (= $SWR/SWR_o$):

$$UVI_{est} = UVI_o \left[ b_0 + b_1 \cdot CMF_{SWR} + b_2 \cdot CMF_{SWR}^2 \right] \tag{3}$$

An identity test using Dummy variables in polynomial equations was used to verify the possibility of fitting a single equation based on the both French and Brazilian databases. For that, dummy variables assumed binary values (0 or 1) for each region. The hypothesis in consideration is $H_0$: $b_{11} = b_{12} = b_k$ (equations are identical) vs. $H_1$: $b_{11} \neq b_{12} \neq b_k$ (equations are not identical) for the complete equation extracted from the equation (2):

$$\frac{UVI_{est}}{UVI_o} = b_{11}D1CMF_x + b_{12}D2CMF_x + b_{21}D1CMF_x^2 + b_{22}D2CMF_x^2 \tag{4}$$

The lower-case type x indicates that the test was performed for PAR and SWR related equations (eqs. 2 and 3, respectively). D1 = 1 for Paris database (or 0 otherwise); D2 = 1 for São Paulo database (or 0 otherwise). The null hypothesis was tested using F-test of analysis of variance (ANOVA) for reduction at 5% (Neter et al., 1996; Gujarati, 2004). The comparison between UVI estimates provided by the standard equation and measurements is showed in the next section.

Finally, Chilean measurements were used to test our method as an independent data series.

## 3. Results and discussions

First of all, we highlight the differences between solar UVR availability in Paris and São Paulo in table 2:





**Table 2.** UV statistics measurements (mean ± standard deviation) performed in Paris (2012-2013) and São Paulo (2005-2009)

| | Daily erythemal dose Jm$^{-2}$ | | | Mean UVI at noon | | |
|---|---|---|---|---|---|---|
| | annual | summer | winter | annual | summer | winter |
| Paris | 2501 ± 1085 | 3035 ± 926 | 209 ± 108 | 2.9 ± 2.3 | 4.7 ± 2.1 | 0.5 ± 0.3 |
| São Paulo | 3381 ± 1529 | 4690 ± 1424 | 2011 ± 578 | 6.1 ± 3.4 | 7.6 ± 4.2 | 4.0 ± 1.6 |

São Paulo is a subtropical site where the UVR fluxes are much higher than in Paris. Brazilian ministry of health data has indicated that large solar UVR availability in Brazil is the main factor responsible for one of the highest rates of skin cancer cases worldwide (32.6% of the all new cancers in 2014). On the other side, the lack of UVR in Paris may prevent the adequate human vitamin D synthesis mainly in winter. Therefore, wider UVR monitoring networks are essential for the public information and health-related studies. In the next sections, we show that a single equation can provide good estimates of UVR radiation from PAR measurements, independently of the site of measurements.

**3.1 Testing the hypothesis for a single equation for UVI estimates from PAR measurements**

Figure 1 shows the scatterplot between CMF$_{UVI}$ and CMF$_{PAR}$ estimates, where red and blue dots represent Paris and São Paulo measurements, respectively. The 2$^{nd}$ degree regression curves are also showed with the same colours. Black line is the regression equation considering the set of the both data sets together.

In order to avoid large instrumental errors, we have only considered UVI and PAR measurements performed under SZA < 60°. Thus, we maintain a significant sample without triggering trends or biases.





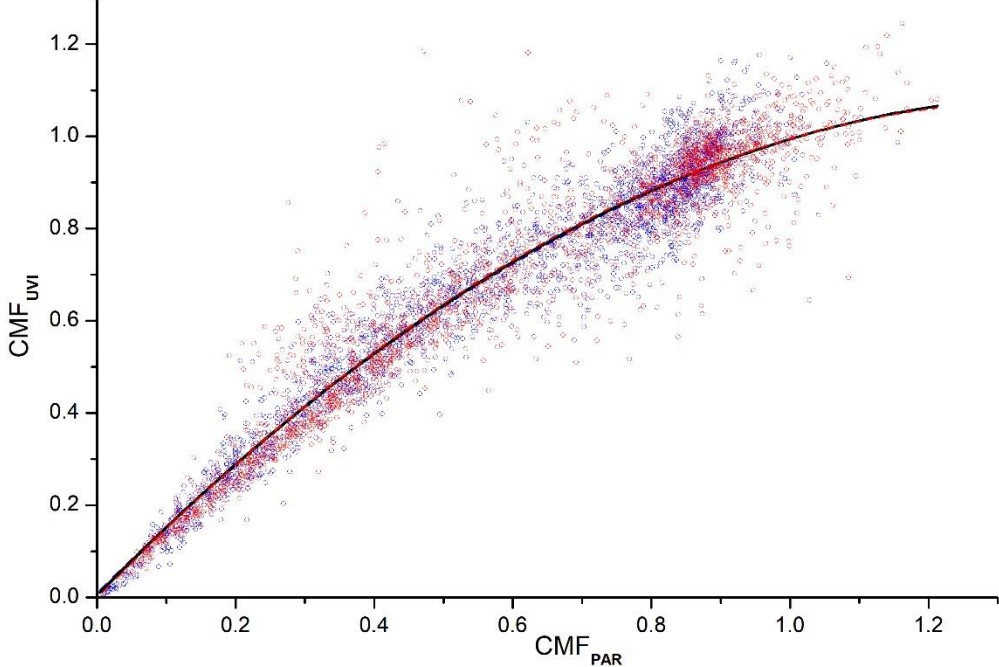

**Figure 1:** Scatterplot of $CMF_{PAR}$ vs. $CMF_{UVI}$ samples. Blue and red dots show São Paulo and Paris data, respectively. Blue and red dashed lines are the 2nd degree regression equation for these sites, respectively. Black solid line is the 2nd degree regression equation for all data (São Paulo and Paris).

The coefficients of the regression equation (eq. 1), standard errors, and the adjusted R-square of the Figure 1 are showed in table 3. Three regression equations are compared in this table.

**Table 3.** Coefficients of the regression equation (eq. 1), standard errors, and the adjusted R-square of the Figure 1 curves

|  | $b_0$ | SE ($b_0$) | $b_1$ | SE ($b_1$) | $b_2$ | SE ($b_2$) | $r^2$ |
|---|---|---|---|---|---|---|---|
| Paris | (-) 0.00 | 0.01 | 1.56 | 0.03 | - 0.56 | 0.02 | 0.92 |
| São Paulo | 0.01 | 0.01 | 1.50 | 0.03 | - 0.51 | 0.02 | 0.94 |
| All data | 0.00 | 0.00 | 1.53 | 0.02 | - 0.54 | 0.02 | 0.93 |

P-values associated with the standard errors show that $b_1$ and $b_2$ coefficients are significantly different from zero ($p < 0.001$). On the other hand, p-values associated with b0 coefficients indicate that this coefficient can be approximate to zero in the three datasets (p = 0.77, 0.12 and 0.34, for Paris, São Paulo and All dataset, respectively).

We test the hypothesis to fit a single equation based on French and Brazilian databases using Dummy variables technique.

10 Table 4 shows that the null hypothesis is not rejected at 5% level of significance, which enable the fitting of a single equation for Paris and São Paulo databases ($H_0$: $b_{11} = b_{12} = b_k$, thus the three equations can be considered similar).



**Table 4.** ANOVA for Dummy variables in polynomial regression models based on PAR measurements

| Source of variance | DF | Mean squares | F-value | p-value |
|---|---|---|---|---|
| Complete Model | 6 | 418.5262 | | |
| Reduced model | 2 | 1255.5707 | | |
| Reduction ($H_0$) | 4 | 0.0033 | 0.6289 | 0.6419* |
| Error | 4680 | 0.0053 | | |

### 3.2. Testing the single equation for UVI estimate from PAR measurements

Now we test the equation $UVI_{est} = UVI_o \left[ 1.53 CMF_{PAR} - 0.54 CMF_{PAR}^2 \right]$ to reproduce the UVI data from previous PAR

5    measurements. Figure 2 shows the absolute differences between UVI measurements performed by an UV radiometer and the

UVI estimates provided by the single equation.

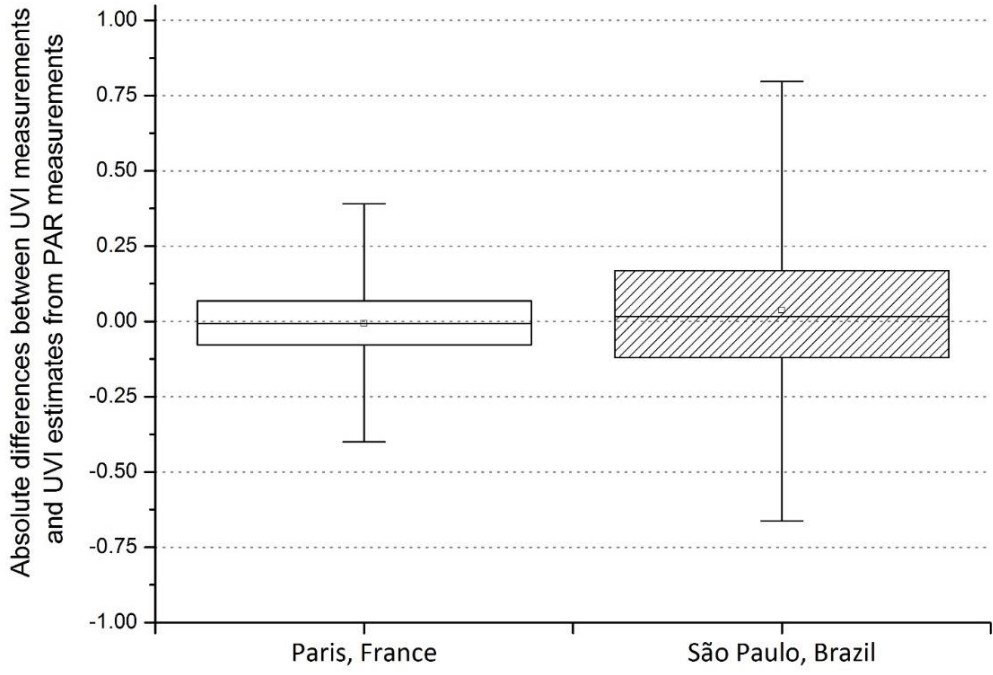

**Figure 2.** Boxplots of the absolute differences between UVI estimates from PAR measurements and regular UVI measurements performed in France and São Paulo. Box horizontal borders represent 1st and 3rd quartiles and the middle horizontal line is the median. Whiskers are the 5th an 95th percentiles, and the small square is the mean of the differences between measurements and estimates.



This simple method provided a significant accuracy for UVI estimates and high correlation with the measurements. 90% of the differences ($P_5$ and $P_{95}$) between UVI estimates and measurements are between -0.7 and 0.8 UVI for São Paulo and between -0.4 and 0.4 for Paris. Interquartile range is about 0.3 and 0.2 for these cities, respectively. Mean relative differences are about 7.5% for São Paulo and -0.5% for Paris. The larger variation observed in São Paulo dataset are related to the

5   greater variations of the UVI during the year and probably due to differences on the cloud cover variability.

Despite the good results for the UVI inference, this method provides better accuracy for UV erythemal daily doses estimates. Relative errors below 10% were observed in the most (95% confidence interval for the means: $-59.1 \, (-2.0\%) < \mu_{differences} < -32.8 \, Jm^{-2} \, (-1.0\%)$), as we show in Figure 3.

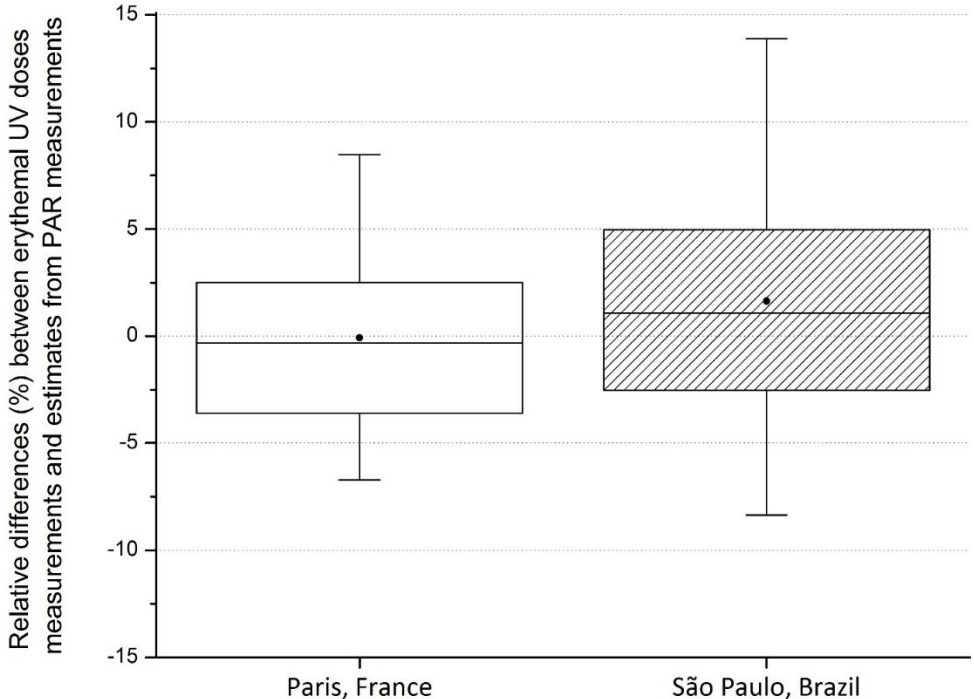

**Figure 3.** Boxplots of the relative differences between daily doses estimates and measurements at the Paris and São Paulo sites. Black circles show mean values, and the whiskers indicate the 5th and 95th percentiles.

10   **3.3. Testing the hypothesis for a single equation for UVI estimates from SWR measurements**

Figure 4 is similar as the Figure 1. It shows the scatterplot between $CMF_{UVI}$ versus $CMF_{SWR}$ estimates and the 2nd degree regression curves for the French and Brazilian sets of data. In this case, we also have considered only measurements performed under $SZA < 60°$.





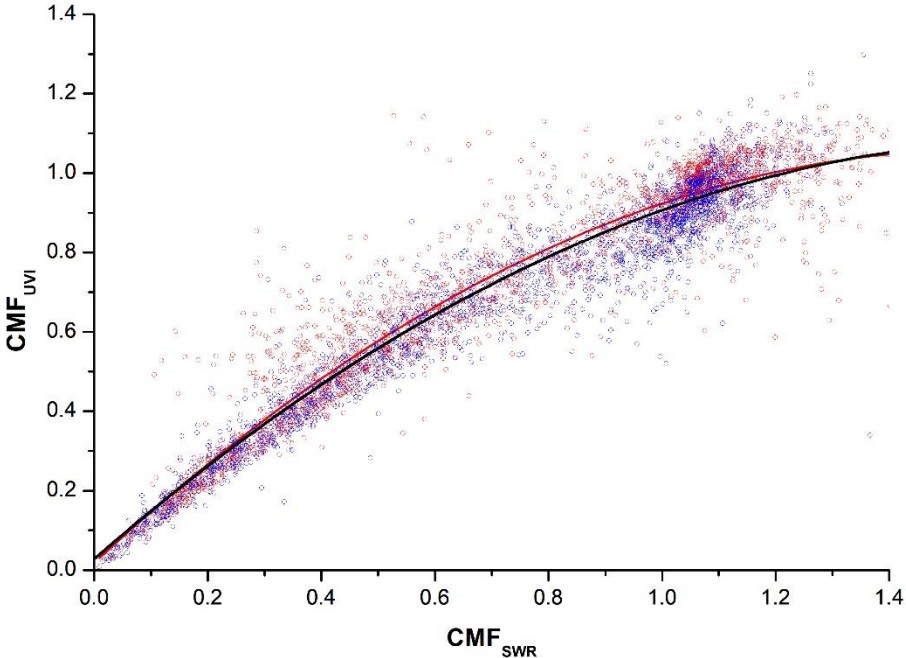

**Figure 4.** Scatterplot of $CMF_{SWR}$ vs. $CMF_{UVI}$ samples (inverse). Blue and red dots show São Paulo and Paris data, respectively. Blue and red dashed lines are the 2nd degree regression equation for these sites, respectively. Black solid line is the 2nd degree regression equation for all data (São Paulo and Paris).

The table 5 shows the coefficients of the regression equation (eq. 2), standard errors, and the adjusted R-square of the Figure 4 curves.

**Table 5.** Coefficients of the regression equation (eq. 2), standard errors, and the adjusted R-square of the Figure 1 curves

|  | $b_0$ | SE ($b_0$) | $b_1$ | SE ($b_1$) | $b_2$ | SE ($b_2$) | $r^2$ |
|---|---|---|---|---|---|---|---|
| Paris | 0.02 | 0.01 | 1.33 | 0.02 | - 0.43 | 0.02 | 0.91 |
| São Paulo | 0.04 | 0.01 | 1.16 | 0.02 | - 0.30 | 0.02 | 0.94 |
| All data | 0.02 | 0.00 | 1.24 | 0.02 | - 0.37 | 0.01 | 0.92 |

P-values associated with the standard errors show that $b_0$, $b_1$ and $b_2$ coefficients are significantly different from zero ($p < 0.01$).

Finally, we test the hypothesis to fit a single equation based on French and Brazilian databases. In this case the null hypothesis is rejected at 5% level of significance ($p < 0.05$), which does not enable the fitting of a single equation for Paris

10 and São Paulo databases. The statistical test shows that despite the similarities among these equations, it is not possible that a unique and simple equation provides accurate UVI estimates from SWR measurements. Near-infrared wavelengths of the SWR spectral range are more sensible to the water vapor and cloud amounts variability (Yamanouchi and Tanaka, 1985).



Therefore, probably the differences between Paris and São Paulo atmospheric conditions contribute to these significant side effects in the SWR data.

**Table 6.** ANOVA for Dummy variables in polynomial regression models based on SWR measurements

| Source of variance | DF | Mean squares | F-value | p-value |
|---|---|---|---|---|
| Complete Model | 6 | 426.2496 | | |
| Reduced model | 2 | 160.8396 | | |
| Reduction (H0) | 4 | 558.9545 | 9544.78 | 0.0005* |
| Error | 4554 | 0.0058 | | |

5   **3.4. Applying the equation for UV estimates from PAR measurements**

A dataset collected in Arica, Chile (see table 1), was used for testing out method for inference of UV radiation from PAR measurements. Figure 5 shows the relative differences between measurements and estimates of UVI and erythemal doses collected in 2011.

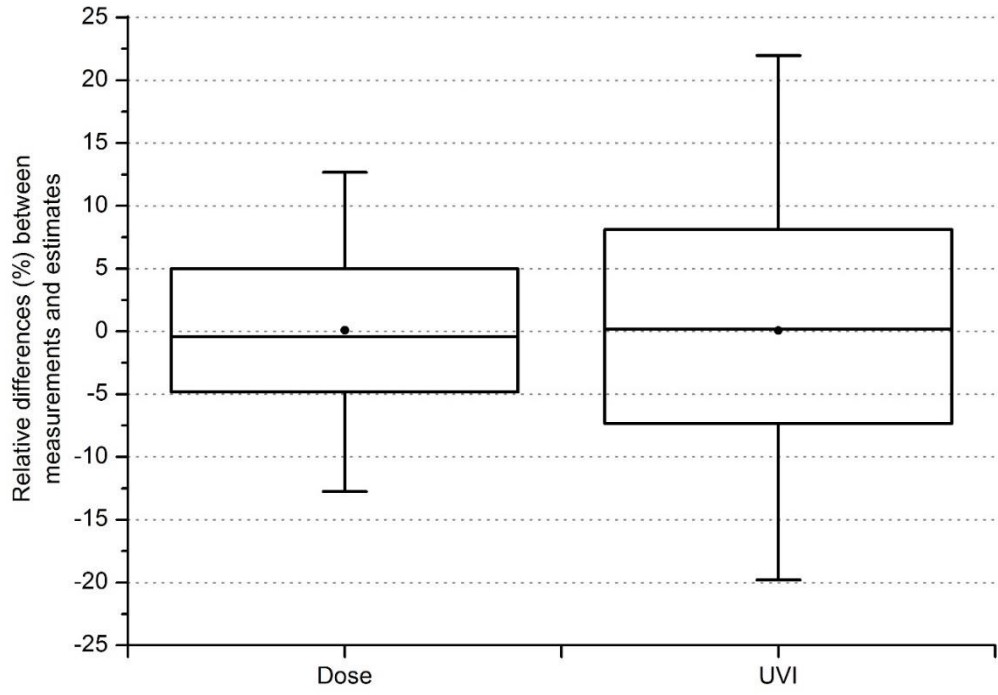

**Figure 5.** Boxplots of the relative differences between estimates and measurements in Chile. Black circles show mean values, and the whiskers indicate the 5th and 95th percentiles.



We are 95% confident that the absolute mean erythemal dose difference is within the range -5 and 91 Jm$^{-2}$, while for the UVI we have differences between 0 and 0.1 UVI. These results are similar as the obtained for Paris and São Paulo. Thus, this simple method seems to provide accurate UV estimates using PAR measurements and a radiative transfer code.

## 4 Conclusions

We have presented a simple method based on a 2$^{nd}$ degree polynomial regression for providing UVI and erythemal doses estimates from PAR measurements and radiative transfer calculations. Our method is based on large solar radiation databases collected in São Paulo (Brazil) and Paris (France) and CMF calculations regarding clear-sky conditions. A statistical identity test using Dummy variables showed that a single equation using the both databases together can provide similar results with good accuracy. We reproduce the Brazilian and French UV data series with high precision (UVI and

erythemal doses differences < 5%). Finally, we also tested the method to reproduce an UV radiation measurements collected in Arica (Chile). Once again, the results were satisfactory showing absolute differences around ± 0.5 UVI in most of the them. We also tested a similar method for providing UV data from SWR measurements, but we did not find statistical support for an equation based on this larger spectral range. We conclude that the equation based on PAR measurements provides accurate results for UV estimates due to the nature of the atmospheric interactions. That is, weak absorption and

strong scattering well represented in radiative transfer models in this visible spectral range. Besides, ozone total content is well estimated by satellite measurements and the absorption in the UV range is also well represented in these models. On the other hand, SWR calculations depends on the atmospheric water vapor absorption in the infrared range. For this reason, an UV-estimate model based on SWR range depends on water vapor measurements and absorption calculations. Therefore, PAR measurements can provide an easy-to-use method for UV inference.

*Competing interests.* The authors declare that they have no conflict of interest.

*Acknowledgements*. Marcelo de Paula Corrêa thanks Fapemig and CNPq for the financial support. Miguel Rivas and Elisa Rojas, wish to thank to the funding provided by the project UTA-Mayor 4730-17.

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
