# Peer review of "Retrieval of health-related UV doses from PAR measurements"

_Atmospheric Measurement Techniques, 2017_

## Referee Comment (RC1) · Anonymous Referee #1 · 15 Mar 2018

Retrieval of health-related UV doses from PAR measurements

Summary The manuscript starts from an interesting premise, namely using PAR and an analogue for UV dose and index measurements, but fails to meet general criteria for acceptance. While it provides a relationship between PAR and UVI and exposure (dose), it states that the agreement has 'good accuracy', but there is no indication in the manuscript that it has met any user requirement of what defines 'good' except their own, and there is no comparison to other predictive index and doses studies using an analogue. The English is poor throughout the article and on every page if not paragraph. This is most disappointing as the authors have highlighted a potential methodology for inferring UVI and daily exposure from PAR. Only after a significant revision should the manuscript be reconsidered for publication.

[Figure]

General Comments 1. One of the reasons that E-UVR measurements are expensive is that they need to be traceable, just like the PAR and SWR measurements. While the purchase cost of a PAR instrument is low, the cost to maintain calibration to a traceable quantity could be as high as that for UVR. 2. There is no indication of the uncertainty (a quantitative parameter) of the measurements used for the paper. Table 1 provides an indication of the instrumentation models but not the uncertainty of each data sample. Given that the instrumentation for SWR is a LI-200R it could be expected that the uncertainties are not those of a high-quality instrument, and there are no references cited for the uncertainties of the multiple measurands of the three data sets. Hence the manuscript is deriving parameters from numbers not traceable measurements. 3. In Table 1 or the text there is no indication of the quantity (irradiance or exposure) measured for each of the data sets, just the 'Data record'. Are they exposures (integration over the time period) or irradiances (average over the time period). Were they under sampled i.e. were they measurements taken every time period, or sampled at a higher rate than the time constants of the instruments? The authors need to demonstrate that the data they are using are traceable measurements. 4. The hypothesis testing of UVI only used data when SZA<60, and at solar noon(?), so the data availability figures in Table 1 are not reflective of actual number used. For example, at a maximum <600 solar noon UVI measurements could be used for Paris if the SZA was never less than 60 deg, but perhaps a 1/3 of a year SZA at noon > 60. The number of points used for each CMFUVI estimate need to be available. 5. Table 4 is largely irrelevant. 6. Figure 2 caption: How can 'absolute' differences be both positive and negative! These are just differences in the UVI. The significance of the range of the 1st and 3rd quartiles is irrelevant without user requirements. Similarly, the use of the term 'significant accuracy', remembering that 'accuracy' is a qualitative not quantitative term, is meaningless to the this reader. 7. Page 8 line 6: 'provides better accuracy' – see point 6 above. 8. There is scant discussion of the exposure (dose) estimation and basically only covers 2 thirds of a page, and the scales are relative rather than in exposure (or dose) units. To be consistent with Figure 2, Figure 3 should be in exposure units. 9. Figure 4 shows that

a majority of clear skies in both Paris and Sao Paulo give CMF>1, and provides this reader with a high degree of scepticism that the data sets of SWR are of any value or the SWR model parameter used were of any value. Again, the number of data points used for each fit would be useful.

Specific comments

10. There are too many errors of grammar and syntax and would almost be the length of the paper. It is suggested that any future version of the
* * *

---

## Referee Comment (RC2) · Anonymous Referee #2 · 15 Apr 2018

This paper suggests an empirical relationship between UVR and PAR measurements, and applies the second order polynomial found to estimate UVR daily doses (and maximum daily UVI) from PAR measured data. Three datasets are used, namely for Paris, France, Rio, Brazil, and Santiago, Chile.

The content of the paper is correct as the data analysis is adequate, and the statistical tests are suitable for the purpose of the paper. It is also correct using two datasets for finding the empirical relation and the third dataset, for validating its performance.

However, there is nothing really new in this study. There are no new instruments, the measurements are analyzed in a quite traditional way, apparently more importance is given to the statistical analyses (see for example the ANOVA tables of results, or sentences as in lines 6-7 of page 9) than to the physical interpretation, and in fact the

justification of the interest of the derived relationship is somewhat poor (see below). In addition, the wording and figures need also a significant improvement. Therefore, I'm sorry to say that in my opinion this paper is not suitable to be published in AMT, at least in its current format.

Next, I will be giving some details about these criticisms, and also I will suggest changes that the authors may wish to consider to include for future versions of this study, either in this or in a different journal. One option would be to further synthetize and present the study as a "short note" or similar.

- The authors insist that this study is important as a way to estimate UVR (since the measurements of UVR are relatively expensive) from PAR measurements (which are performed by cheaper instruments). While this is true, it is not right to say that PAR measurements are performed at most meteorological stations. To my knowledge, if a meteo station has a radiometer, it is usually a pyranometer, that is an instrument that measures the whole solar radiation band (0.4-4 microns aprox.) Only a few stations (in particular, those devoted to agrometeorology) are equipped with quantum sensors for PAR measurements. In summary, I have some doubts about the practical interest and applicability of the obtained expression.

- The authors give so little details about the use of the radiative transfer models to estimate the "clear" sky radiation which is necessary for computing the CMFs. They say that two models have been used, but does this mean that one has been used for PAR and the other for UVR? What about the estimations for SWR? How did the models where applied (spectral resolution, fixed parameters, postprocessing of model results,...)

- Related with the above two points, I would suggest a more "physically oriented" paper, that would explore (by combining measurements with modeling results) the differences between using SWR or PAR for estimating UVR. Although the authors already mention these differences (notably, the water vapor absorption that does affect SWR), this could

be further explored, explained and shown.

Other, minor questions:

p 2, l 3. I wouldn't say that UVR instruments, even if low cost, can be found at "ordinary shops".

p 2, l 14-16. The two sentences are somewhat contradictory. Could you at least add some additional reference here to support the "simple cloud transmission functions"?

P 2, l25. There are already sky imagers that are not "very expensive". I would say that there are imagers at the same or similar cost as a UV radiometer or a good pyranometer.

Table 1: please set clearly that the UV instrument is for "erythemal UV"

Eq 2, 3, 4. As you have already defined CMF, you could use CMF in these equations, instead of writing UVI_est and UVI_0.

Table 2. Which months are considered summer and winter months?

p 5, l 9. You don't need to write "UVR radiation" as R already refers to radiation

p 5, l 11. Figure 1 shows measurements, not "estimates".

p 5, l 14-15. The limitation to SZA < 60 deg should be mentioned in the Methods question. Moreover, the sentence "we maintain a significant...biases" should be better justified and explained.

Fig 1. The plot is of CMF_UVI vs CMF_PAR. The blue and red lines at not visible, so the caption should mention that these lines cannot be observed because the are virtually the same, and the same as the black line.

p 6, l 1-2, and caption Table 3. The coefficients correspond to Eq. (2), not (1).

p 6, l 7-8. It is not surprising that coefficient b_0 is null. Actually, this is what we should expect from physical reasoning, isn't it?

Sections 3.2-3.4, and Fig. 2-3-5. You should make clear which differences are you referring to: estimates – measurements (as it should be) or measurements – estimates (as it is stated in the text and in the axis labels). Probably, Fig. 2-3 could be joint in a single, two-panel figure.

p. 11, l 14-15. I don't understand that the result obtained in this study come from the fact that absorption and scattering is well represented in radiative transfer models.